# The Association Between *Prevotella copri* and Advanced Fibrosis in the Progression of Metabolic Dysfunction-Associated Steatotic Liver Disease

**DOI:** 10.3390/nu17132145

**Published:** 2025-06-27

**Authors:** David Zhang, Madelaine Leitman, Shrey Pawar, Simer Shera, Laura Hernandez, Jonathan P. Jacobs, Tien S. Dong

**Affiliations:** 1Department of Medicine, University of California, Los Angeles, CA 90095, USA; dzhang2026@g.ucla.edu (D.Z.); madleitman2003@g.ucla.edu (M.L.); sshera2024@g.ucla.edu (S.S.); jjacobs@mednet.ucla.edu (J.P.J.); 2Vatche and Tamar Manoukian Division of Digestive Diseases, Department of Medicine, University of California, Los Angeles, CA 90095, USA; 3Goodman-Luskin Microbiome Center, University of California, Los Angeles, CA 90095, USA

**Keywords:** *Prevotella copri*, MASLD, advanced fibrosis, gut microbiome, hepatic steatosis, lipid metabolism

## Abstract

**Background/Objectives**: Metabolic dysfunction-associated steatotic liver disease (MASLD), driven by obesity and metabolic syndrome, is increasingly prevalent and a significant contributor to liver fibrosis, cirrhosis, and liver-related mortality. Emerging research implicates the gut microbiome as a critical player in MASLD progression, yet specific microbial drivers remain poorly understood. Here, we explore the role of *Prevotella copri* (*P. copri*) in MASLD progression through both human patient cohorts and a mouse model of diet-induced obesity. **Methods/Results**: Using 16S rRNA sequencing, we identified elevated *P. copri* abundance in MASLD patients with advanced fibrosis, linked with significant shifts in microbial diversity and bacterial network connectivity. To investigate causality, experimental colonization of *P. copri* in mice on a high-fat diet worsened MASLD progression, with *P. copri*-colonized mice showing significant increases in hepatic steatosis, liver triglyceride accumulation, and body weight, independent of caloric intake. At the molecular level, *P. copri* colonization downregulated key lipid metabolism genes, such as carnitine palmitoyltransferase 1 and adipose triglyceride lipase, and impaired tight intestinal junction integrity through the downregulation of occludin. Collectively, our findings position *P. copri* as a possible driver of MASLD progression by promoting hepatic steatosis through lipid and triglyceride accumulation and fibrosis through decreased tight junction integrity. These insights suggest a promising therapeutic avenue to target specific microbial signatures like *P. copri* to curb MASLD progression and mitigate the associated risk of advanced fibrosis.

## 1. Introduction

Metabolic dysfunction-associated steatotic liver disease (MASLD), formerly known as non-alcoholic fatty liver disease, is now the most common cause of liver disease worldwide, affecting over 30% of adults [1]. The dramatic rise in MASLD prevalence—from 25.3% in the early 1990s to 38.2% in recent years—closely mirrors the global increase in obesity. Obesity is a primary driver of MASLD, as excess adipose tissue contributes to insulin resistance, systemic inflammation, and increased hepatic fat accumulation. These metabolic disturbances directly promote the onset and progression of liver steatosis. As a result, MASLD has emerged as a major consequence of the obesity epidemic, with approximately 20% of individuals progressing to cirrhosis over time, often followed by hepatocellular carcinoma and increased mortality risk [2,3,4,5].

Healthy liver function is essential for numerous physiological processes, including metabolism, detoxification of harmful substances, bile production for digestion, and maintenance of immune homeostasis. Approximately 80% of the hepatic blood supply is delivered from the gut through the portal vein [6]. This ensures that nutrients absorbed from food, as well as potential toxins, pathogens, and waste products, are processed by the liver before reaching the rest of the body. The intricate crosstalk between the gut and liver via the gut–liver axis is not only through the portal vein but also through gut permeability [7], lymphatic vessels [8], gut-derived lipid messengers [9], and vagal sensory afferent nerves [10].

Given these major interactions between the gastrointestinal tract and the liver, it is not surprising that the gut microbiome has been identified as an important factor in the pathogenesis of liver disease, including MASLD. Prior research has found that patients with liver cirrhosis have impaired gut-barrier function, with a significant compositional shift towards pro-inflammatory taxa [11]. This unbalanced state known as dysbiosis has a critical role in short-chain fatty acid (SCFA) production and altered intestinal permeability [11,12]. Failure to clear ‘dangerous’ stimuli from the gut leads to pathological inflammation and disrupted tissue homeostasis characterized by the development of fibrosis [13].

While MASLD has been associated with increased risk of cardiovascular events, the strongest predictor of liver-related mortality is the development of liver fibrosis [2,14,15]. Hepatic inflammation can only accurately be detected through liver histology, which is not practical for widespread use due to its invasive nature. Consequently, the early diagnosis of advanced fibrosis through non-invasive methods has become a critical focus in hepatology research [13,14]. However, our understanding of the risk factors associated with liver fibrosis progression is still under investigation.

Prior research from our laboratory demonstrated that microbial profiles could accurately predict advanced fibrosis in chronic liver disease, with the highest predictive power observed for MASLD [16]. In this study, we aim to uncover microbial signatures that play a role in the progression of MASLD, then subsequently validate these associations through rigorous experimental models. Our findings underscore the potential of the gut microbiome as a target for early diagnostic and therapeutic strategies in MASLD.

## 2. Materials and Methods

### 2.1. Patient Recruitment and Stool Collection

Patients diagnosed with MASLD who were undergoing ultrasound elastography were prospectively recruited from the Veterans Affair Greater Los Angeles Healthcare System (VA) between June 2017 and June 2018. Exclusion criteria included recent antibiotic or probiotic use (within 3 months prior to enrollment), isolated acute liver injury without any underlying chronic liver disease, successfully treated hepatitis C virus with sustained virologic response without other liver disease, use of specialized diets (such as gluten-free, vegan, vegetarian, or high-protein diets), prior gastrointestinal surgeries, irritable bowel syndrome, or inflammatory bowel disease. Stool samples were collected within 7 days of the ultrasound elastography and preserved in 95% ethanol at −80 °C until analysis. Stool samples from healthy controls without signs of chronic liver disease were also collected for comparison.

Participant demographics and clinical information were also recorded, including age, gender, race/ethnicity, and comorbidities. Racial and ethnic categories included non-Hispanic white, non-Hispanic black, Hispanic, Asian, and other, with Hispanic patients categorized separately. Comorbidities were assessed using the Charlson Comorbidity Index, which provides a validated measure of overall health and 1-year mortality risk. This study received approval from the Institutional Review Board of the VA Greater Los Angeles Healthcare System, and all study methods complied with relevant guidelines and regulations. Informed consent, both written and verbal, was obtained from all participants.

### 2.2. Liver Ultrasound Elastography

In order to assess fibrosis, all patients with MASLD underwent ultrasound elastography using the FibroScan Touch 502 device (Echosens, Westborough, MA, USA). Trained technicians, each with over 100 scans of experience, performed all elastography procedures. Based on the patient’s body habitus, either medium or extra-large probes were used according to the manufacturer’s instructions. Measurements of the controlled attenuation parameter (CAP) score and liver stiffness were collected as non-invasive indicators of hepatic steatosis and fibrosis, respectively.

Each assessment was taken a minimum of 10 times at the same location, with an interquartile range/median value below 30% in accordance with manufacturer guidelines. The CAP score was interpreted as follows: scores between 238 and 260 were classified as grade S1, indicating 11–33% fatty change in the liver; scores from 260 to 290 were classified as grade S2, representing 34–66% fatty infiltration; and scores above 290 corresponded to grade S3, indicating more than 67% fatty change, as defined by the manufacturer. Standard liver stiffness cutoffs, measured in kilopascals (kPa), were used to stage fibrosis from F0/F1 (minimal fibrosis) to F3/F4 (advanced fibrosis). Consistent with previous studies, minimal fibrosis was defined as scores indicative of F0–F2, while advanced fibrosis corresponded to F3–F4.

### 2.3. 16s rRNA Sequencing

DNA was extracted from stool samples preserved in ethanol using the Powersoil DNA Isolation Kit following the manufacturer’s protocol (MO BIO, Carlsbad, CA, USA). Amplification targeted the V4 region of the 16S rRNA gene, and paired-end sequencing was performed on an Illumina HiSeq 2500 platform (San Diego, CA, USA), as described in previous studies. Each 253 base-pair read was processed using the DADA2 pipeline in QIIME 2 version 2024.5 (San Diego, CA, USA), which denoises reads to generate amplicon sequence variants (ASVs) at a single-nucleotide resolution. Sequencing yielded an average depth of 35,613 sequences per sample. Taxonomic classification of ASVs was performed using the SILVA database (release 138), ensuring accurate and up-to-date taxonomic assignments. To reduce noise from low-abundance taxa, ASVs with a frequency below 0.0001 (178 reads) were filtered out. The raw 16S rRNA sequencing data were submitted to the National Center for Biotechnology Information under BioProject PRJNA542724.

### 2.4. Patient Metadata Analysis

To analyze demographic characteristics across groups, we performed statistical tests to compare age, gender distribution, comorbidities (using the Charlson Comorbidity Index), and race across healthy controls, MASLD patients without advanced fibrosis, and MASLD patients with advanced fibrosis. Specifically, we calculated the mean and standard deviation for each continuous variable, then used the Kruskal–Wallis test. For categorical variables, we summarized groups with a contingency table to show the distribution of each category. Fisher’s exact test was used to assess whether there were significant differences in group distributions.

### 2.5. Diversity Analysis

Microbial diversity analyses were carried out in QIIME2 (version 2024.5) [17] within an Anaconda environment, which is a virtual workspace that isolates specific versions of Python (version 3.13.5) and required software packages. To allow for balanced comparisons across all samples, alpha diversity was assessed at a rarefaction depth of 5311, ensuring the inclusion of all samples while maintaining adequate sequencing coverage. Alpha diversity indices, including the Shannon diversity index, Chao1 species richness index, dominance index, and abundance-based coverage estimator (ACE) index, were calculated using the qiime diversity core-metrics command. For beta diversity, metrics such as the robust Aitchison distance were also computed with the core metrics command. After the diversity metrics were generated, they were exported from QIIME2 for further statistical analysis in R (version 4.4.1). Alpha diversity comparisons were performed using the Mann–Whitney U test via the wilcox.test function, and beta diversity differences were evaluated using Permutational Multivariate Analysis of Variance (PERMANOVA) from the vegan package (version 2.7). A significance level of 0.01 was applied for all tests. Visualizations of diversity metrics were created in R using ggplot2.

### 2.6. Bacterial Co-Occurrence Network

To investigate microbial associations and differences in microbial interactions between healthy controls, MASLD patients without advanced fibrosis, and MASLD patients with advanced fibrosis. The Sparse Co-Occurrence Network Investigation for Compositional Data (SCNIC) (version 0.1) tool within the QIIME2 environment was employed for network construction. Relationships between taxa were calculated using Sparse Correlations for Compositional data (SParCC), based on filtered count tables for each group.

Correlations of greater than or equal to 0.5 in magnitude (r ≥ |0.5|) were imported into Cytoscape (version 3.10.3) for visualization and nodes were colored by Phylum. Negative correlations are distinguished from positive by a zig-zagged edge connection. To evaluate correlation between the co-occurrence dissimilarity matrices, we conducted a Mantel test using the mantel function in the Vegan library in R. The test was based on Pearson correlation (method = ‘pearson’) with 999 permutations to determine statistical significance, yielding a correlation coefficient and *p*-value.

### 2.7. Differential Taxonomic Abundance Analysis

For differential taxonomic abundance analysis, we utilized MaAsLin2 (Multivariate Association with Linear Models), a tool developed for identifying associations between microbial features and metadata variables. The feature table and metadata were imported into R, where MaAsLin2 (version 3.21) was used to perform the analysis. To account for compositional effects, the data was normalized using total sum scaling (TSS), followed by a log transformation to improve interpretability and meet model assumptions. Advanced fibrosis status was set as a fixed effect to evaluate its association with microbial abundance. We used a linear model (LM) as the analysis method, and associations were deemed significant at a q-value threshold of 0.1. Only taxa with q < 0.1 were considered significantly associated and were visualized in subsequent plots to highlight bacteria with differential relative abundance.

### 2.8. Mice

To explore the role of *P. copri* in metabolic-associated fatty liver disease (MASLD), we employed a diet-induced obesity model in a cohort of 32 C57BL/6 mice sourced from Jackson Laboratory. Mice were housed in a specific pathogen-free environment and maintained on a 12 h light/dark cycle. Only male mice were used, as male mice develop more MASLD than female mice. All animal protocols adhered to the guidelines and regulations set by the institution. There were a total of 4 groups with 8 mice in each group. Average baseline weights were similar in each group.

The mice were randomly assigned to either a standard control diet or a high-fat (HF) diet (40% fat, 2% cholesterol, and 20% fructose) to induce obesity and liver disease phenotypes. In summary, 16 mice received a HF diet, which we called the HF group, and 16 mice received a standard diet, which we called the ‘Control’ group.

### 2.9. Oral Gavage

After assigning diets, half of the mice in each diet group were selected to receive *P. copri* inoculation to evaluate its impact on MASLD. These mice were orally gavaged with *P. copri* at a concentration of 1 × 10^9^ colony forming unit per 200 µL gavage, while the remaining mice received a vehicle control (sterile phosphate-buffered saline). For the first four weeks, inoculations were administered twice weekly; afterward, the gavage continued once weekly, maintaining this schedule for a total of 96 days. Before each gavage, mice weights and food intake were recorded. The inoculation approach allowed for continuous colonization and exposure to *P. copri* throughout the study period. After this experimental period, the mice were sacrificed for future assays.

### 2.10. Hematoxylin and Eosin (H&E) Staining

Liver tissue samples were collected from all experimental mice at the end of the 16-week study period for histological analysis. Samples were fixed in 10% neutral-buffered formalin, dehydrated, and embedded in paraffin. Thin sections (5 μm) were cut, mounted onto glass slides, and subjected to standard hematoxylin and eosin (H&E) staining. Stained sections were examined under a light microscope to assess histological changes associated with NAFLD, including hepatocyte ballooning, inflammation, and fibrosis.

To quantify liver fat content, stained sections were analyzed using ImageJ software (version 1.54p). High-resolution images of each stained section were captured, and the percentage of fat area was calculated by thresholding to distinguish lipid droplets within hepatocytes from the surrounding tissue. The quantification was performed by measuring the area occupied by fat droplets relative to the total tissue area in each image. This approach provided a quantitative assessment of liver fat accumulation across experimental groups.

### 2.11. Hepatic Triglyceride Quantification

Liver triglyceride content was assessed using the Abcam Triglyceride Quantification Kit (Abcam, Cambridge, UK, ab65336). Liver tissue samples were homogenized in isopropanol, and the lipid extracts were clarified by centrifugation, then the supernatant was used for analysis following the kit’s protocol. Absorbance was measured at 570 nm using a microplate reader, and triglyceride concentrations were determined. Results were normalized to tissue weight and expressed as mg of triglyceride per gram of liver tissue.

### 2.12. Quantitative PCR (qPCR)

To assess the expression levels of genes involved in lipid metabolism and tight junction integrity, qPCR was performed on liver tissue samples. Genes analyzed included carnitine palmitoyltransferase 1 (*Cpt1*), diacylglycerol acyltransferase (*Dgat*), and adipose triglyceride lipase (Atgl) for lipid metabolism, as well as claudins (Cldn), occludin (Ocln), and zonula occludens-1 (Zo) for tight junction integrity (primer sequences can be found in the Appendix A). Total RNA was extracted from the genes, and then RNA samples were reverse-transcribed into complementary DNA. qPCR was carried out on a QuantStudio 3 real-time PCR system (Applied Biosystems, Waltham, MA, USA) using PowerUp SYBR Green Master Mix (Thermo Fisher Scientific, Waltham, MA, USA). Gene-specific primers were designed and validated for each target gene. Relative gene expression levels were calculated using the 2^−ΔΔCT^ method, with normalization to glyceraldehyde-3-phosphate dehydrogenase.

## 3. Results

### 3.1. Characterization of MASLD Patients and Healthy Controls

A total of 25 patients with MASLD and 25 healthy controls were recruited (Table 1). Of the patients with MASLD, 18 individuals did not have advanced fibrosis, which was defined as a fibrosis score consistent with F0–F2, and 7 individuals had advanced fibrosis, which was defined as a fibrosis score consistent with F3–F4. The healthy control cohort was younger on average than the patients with MASLD (*p* < 0.001) and comprised more females (*p* = 0.018). There was no significant difference in the Charlson Comorbidity Index between MASLD patients with and without advanced fibrosis (*p* = 0.06). Furthermore, there was no difference in race/ethnicity between any groups (*p* = 0.484).

### 3.2. Disease Status and Fibrosis Stage Influence Microbial Profiles

To evaluate the differences in bacterial communities associated with MASLD progression, we analyzed both alpha and beta diversity. Alpha diversity was assessed using multiple indices: the abundance-based coverage estimator (ACE) index, Chao1 index, Shannon index, and Simpson index (Figure 1A). Significant differences in alpha diversity were observed in the ACE (*p* = 0.011) and Chao1 (*p* = 0.014) indices, both of which estimate richness—the number of distinct species in a sample—with a focus on rare taxa. In contrast, the Shannon and Simpson indices, which measure overall species diversity by accounting for both richness and evenness, showed no significant differences (*p* = 0.375 and *p* = 0.709, respectively).

Age, gender, race, and Charlson’s Comorbidity Index were included in the multivariate analysis, of beta diversity where results confirmed that microbial profiles differed significantly between healthy controls, MASLD patients without advanced fibrosis, and MASLD patients with advanced fibrosis (*p* = 0.001) (Figure 1B). Together, these findings highlight that disease status and fibrosis stage are linked to distinct shifts in microbial community composition.

### 3.3. Disease Status and Fibrosis Stage Influence Bacterial Interactions

In the mammalian intestine, bacteria coexist within complex communities, often relying on other species for essential nutrients required for growth. This interdependence fosters the formation of specific ecological niches, where bacteria interact within structured communities. To examine differences in these microbial associations across health and disease states, we constructed species-species co-occurrence networks for each cohort: healthy controls, MASLD patients without advanced fibrosis, and MASLD patients with advanced fibrosis (Figure 2A–C). Bacterial co-occurrence networks capture potential interactions between species—such as symbiotic, competitive, or neutral relationships—by displaying significant correlations (r > |0.5|) between bacterial taxa.

We observed notable differences in bacterial interactions among the groups. Quantitatively, each group exhibited significant dissimilarity in microbial network structure, as determined by Mantel tests. This statistical approach allowed us to compare the overall similarity between the distance matrices of each group’s bacterial co-occurrence network, providing a rigorous measure of how microbial community interactions differ across the stages of MASLD progression. The following bacterial co-occurrence patterns across disease states were compared: comparisons between healthy controls and MASLD patients without advanced fibrosis, healthy controls and MASLD patients with advanced fibrosis, and MASLD patients with and without advanced fibrosis yielded significant dissimilarity (Mantel statistic r = 0.04805, *p* = 0.001; r = 0.02237, *p* = 0.032, and r = 0.02472, *p* = 0.017, respectively).

Visually, the bacterial co-occurrence network for MASLD patients with advanced fibrosis displayed a much higher degree of interconnectedness between taxa (Figure 2C), with more frequent and stronger associations. This pattern contrasts sharply with the networks observed in the healthy control group (Figure 2A) and the MASLD without advanced fibrosis group (Figure 2B), where microbial interactions appeared more stratified and loosely connected. This disparity can be attributed to the network only including strong correlations (r > |0.5|), which are shown in Figure 2E. In both the healthy control and MASLD without advanced fibrosis networks, bacterial interactions were primarily of lower correlation values (Figure 2D), with high frequency peaks around r = 0, indicating fewer strong associations. In contrast, the MASLD with advanced fibrosis group exhibited pronounced peaks in high-correlation interactions compared to healthy controls and the MASLD without advanced fibrosis group (Figure 2E), with many taxa having correlation values greater than or equal to 0.5. Such closely linked microbial communities may become more susceptible to fluxes and shifts toward dysbiosis, as any perturbation could disrupt these interdependent relationships.

In terms of interaction type, we also noted an increase in negative interactions with the progression to advanced fibrosis. Negative interactions—where the abundance of one species is inversely related to that of another—imply competitive or antagonistic relationships, such as competition for resources or the production of inhibitory compounds. Among significant correlations (Figure 2E), the healthy control network had 35.90% negative interactions, and the MASLD without advanced fibrosis network showed a reduced 20.88% of negative interactions. In stark contrast, the MASLD with advanced fibrosis network displayed 42.92% negative interactions, suggesting an intensified level of competitive inhibition and antagonism between taxa in this advanced disease state.

### 3.4. Disease Status and Fibrosis Stage Influence Phylum- and Genus-Level Microbial Composition

To examine how MASLD progression influences the taxonomic composition of the gut microbiome, we visualized relative abundances of bacterial phyla and genera across groups and performed differential abundance analyses using MaAsLin2 (ver 3.21). At the phylum level, visual inspection indicated notable differences in MASLD patients with advanced fibrosis compared to other groups (Figure 3A). However, statistical analysis using MaAsLin2 did not reveal any significant associations between phylum composition and advanced fibrosis. Interestingly, across all groups, the Bacteroidetes phylum consistently accounted for over 50% of the total microbial composition.

At the genus level, however, significant associations were observed. MaAsLin2 analysis revealed that several genera differed significantly across disease stages, with MASLD progression showing marked shifts in specific bacterial populations (Figure 3B). Figure 3B visualizes genera that made up at least 1% of the total abundance within any single group. From this plot, there was a decrease in relative abundance of the *Bacteroides* genus, while there was an increase in the relative abundance of the *Prevotella 9* genus in the MASLD with advanced fibrosis group.

### 3.5. P. copri Is More Abundant in MASLD Patients with Advanced Fibrosis

MaAsLin2 analysis at the species level identified significant differences in the abundances of several bacterial species across the healthy control, MASLD without advanced fibrosis, and MASLD with advanced fibrosis groups (q < 0.1, with q < 0.01 as significant), as shown in Figure 3C. These species, in order of statistical significance, included *Prevotella copri* (*P. copri*), *Alistipes obesi*, (*g*) *Negativibacillus*, (*g*) *Christensenellaceae R-7 group*, (*g*) *Ruminococcaceae UCG-002*, *Acidaminococcus intestini*, (*g*) *Lachnospira*, *Faecalibacterium CM04 06*, *Bacteroides uniformis*, and (*g*) *Desulfovibrio*. Among these, *P. copri* stood out as the most differentially abundant species (q = 7.68 × 10^−5^), showing a pronounced increase in relative abundance in the MASLD with advanced fibrosis group compared to both other groups. This significant enrichment of *P. copri* suggests a potential role for this species in MASLD progression, motivating us to further investigate its role in MASLD progression experimentally.

### 3.6. P. copri Colonization Increase Mice Weights on a High-Fat Diet

In order to investigate the causality of *P. copri* on MASLD progression, we utilized a mouse model of obesity and MASLD. As expected, mice on a HF diet demonstrated a marked increase in body weight compared to those on a standard control diet, consistent with the obesogenic effects of a HF regimen. The focus of our study was to assess whether *P. copri* colonization would further exacerbate symptoms within the context of a HF diet. Body weight percent change was monitored over 96 days, with *P. copri*-colonized HF mice consistently showing elevated weight gain relative to HF controls at multiple time points (*p* < 0.01) (Figure 4A). Given the significant findings in the HF diet group and its relevance to MASLD pathogenesis, our subsequent analyses focus solely on the HF groups (i.e., HF Control and HF *P. copri*).

Despite the substantial differences in body weight, daily food intake did not differ significantly between HF diet mice with and without *P. copri* colonization (Figure 4B). Both the HF control and HF *P. copri* groups maintained similar levels of food consumption throughout the experimental period, indicating that the observed increase in body weight in the *P. copri*-colonized HF group was not attributable to increased caloric intake.

### 3.7. P. copri Colonization in Mice on a High-Fat Diet Increases Hepatic Lipid Content

Histological examination of liver sections using hematoxylin and eosin (H&E) staining revealed substantial differences in hepatic lipid accumulation across experimental groups (Figure 5A). Mice on a HF diet exhibited increased fat deposits in the liver compared to those on a standard control diet, confirming the obesogenic and steatotic effects of a HFt diet. Notably, HF diet mice colonized with *P. copri* displayed pronounced lipid accumulation within the liver, with larger and more numerous lipid droplets compared to HF control mice, indicating that *P. copri* exacerbates hepatic fat accumulation under HF dietary conditions.

Quantification of liver fat content using ImageJ further substantiated these observations (Figure 5B). HF *P. copri* mice showed a significantly higher percentage of fat content in liver tissue compared to HF control mice (*p* = 0.014), suggesting that *P. copri* colonization contributes to enhanced hepatic steatosis beyond that induced by a HF diet alone. To confirm the increase in hepatic lipid content, liver triglyceride concentrations were measured biochemically. HF *P. copri* mice exhibited significantly elevated liver triglyceride levels compared to HF control mice (*p* = 0.008; Figure 5C).

### 3.8. P. copri Colonization in Mice on a High-Fat Diet Leads to Downregulation of Genes Involved in Lipid Metabolism in the Liver

MASLD is characterized by the accumulation of excess fat within liver cells, and this buildup is primarily driven by imbalances in lipid metabolic processes, including lipogenesis, lipolysis, and fatty acid oxidation. Disruptions in these processes create an environment where fat accumulates in the liver, causing a cascade of harmful effects that contribute to MASLD progression.

To investigate whether lipid metabolism plays a role in the mechanisms by which *P. copri* exacerbates hepatic lipid accumulation, we analyzed the expression of genes involved in lipid metabolism, including *Cpt1*, *Dgat*, and *Atgl* (Figure 5D–F). HF *P. copri* mice exhibited significantly reduced expression of *Cpt1* (*p* = 0.011) and *Atgl* (*p* < 0.001) compared to HF control mice, indicating a suppression of fatty acid oxidation and lipolysis, respectively. No significant difference was observed in the expression of *Dgat* between HF *P. copri* and HF control groups.

### 3.9. P. copri Colonization in Mice on a High-Fat Diet Leads to Downregulation of Genes Involved in Tight Junction Integrity

Tight junctions are protein complexes that connect adjacent epithelial cells, maintaining the integrity of the gut barrier and controlling the passage of substances from the gut lumen into the bloodstream. Disruptions in tight junction integrity here could increase intestinal permeability (often called ‘leaky gut’). To assess the impact of *P. copri* on tight junction integrity, we analyzed the expression of genes associated with tight junction proteins in the hepatic flexure, including Occludin (Ocln), Zonula Occludens (Zo), and Claudin (Cldn) (Figure 6A–C).

Mice on a HF diet colonized with *P. copri* exhibited significantly reduced expression of Ocln (*p* = 0.019) compared to HF control mice. While there was a trend toward reduced expression of Zo (*p* = 0.063) and Cldn (*p* = 0.138) in the HF *P. copri* group compared to HF controls, these differences did not reach statistical significance. Nonetheless, the overall pattern suggests that *P. copri* colonization may negatively impact the expression of multiple tight junction proteins, potentially weakening the gut barrier.

## 4. Discussion

Metabolic dysfunction-associated steatotic liver disease (MASLD) is of growing concern due to its escalating prevalence, which is closely linked to the global surge in obesity, metabolic syndrome, and type 2 diabetes mellitus [18]. While emerging research suggests that the gut microbiome is largely at play in the increasing prevalence, the microbial species and mechanisms of these changes remain unexplored. To fill this gap, we explored the role of *P. copri* in the progression of MASLD in both human and mice cohorts.

In our racially diverse patient cohort, we observed that MASLD progression is marked by significant shifts in microbial composition. The dysbiotic state seen with MASLD progression—as shown by the significantly different alpha and beta diversity metrics—corroborates findings from other studies [19,20]. Dysbiotic trends in disease pathogenesis emphasizes a bidirectional relationship between host genetics, epigenetics, and the gut microbiome. Host genetic and environmental factors shape the gut microbiome’s composition. Meanwhile, the gut microbiota and its metabolites can induce epigenetic modifications in the host, thereby influencing the development and progression of MASLD [21].

To compound the compositional variances, our bacterial co-occurrence network analysis revealed stark differences in bacterial interactions within the gut microbiomes of each group. The healthy and non-advanced fibrosis groups yielded relatively simple networks with weaker associations between taxa, suggesting a balanced microbial community with less interdependence among species. In contrast, the advanced fibrosis group displayed marked increases in both the strength and complexity of microbial interactions. This dense, interconnected network seen in advanced fibrosis may render the microbial community more susceptible to disruptions.

We also observed a substantial increase in negative interactions in the advanced fibrosis group, where the abundance of one species inversely affects the abundance of another. This increase in antagonistic relationships likely reflects a shift toward a more competitive microbial environment, where species may produce pro-inflammatory or inhibitory compounds to outcompete each other. These microbial metabolites can induce epigenetic changes in hepatic and immune cells, promoting the fat accumulation and inflammation seen in advanced fibrosis [22]. This inflammatory cascade aligns with the type 3 inflammatory response characteristic of MASLD, which involves the following three progressive phases: (1) an initial phase in which innate, tissue-resident immune cells detect hepatic stress and produce cytokines, (2) the recruitment and activation of pro-inflammatory cells—primarily of the myeloid lineage—that amplifies these initial signals, and (3) an escalation of inflammation involving adaptive immune cells, leading to widespread tissue damage [22,23].

Beyond overall dysbiosis driving MASLD progression, *Prevotella copri* emerged as the most significantly abundant species in MASLD patients with advanced fibrosis, with drastic increases compared to the other groups. *P. copri*, a common member of the human gut microbiome, belongs to the *Prevotella* genus within the *Bacteroidetes* phylum [24]. While it has been curiously associated with both positive and negative impacts on diseases [24], our findings align with other published work in relation to MASLD, which have linked *P. copri* to MASLD development through lipopolysaccharide biosynthesis and fat accumulation [25,26,27,28]. Other published work has shown that *Prevotella* was enriched in patients with cirrhosis as compared to healthy controls [29], and *P. copri* was the main driver of advanced fibrosis in MASLD pediatric patients [30].

*P. copri* has been extensively studied in other inflammatory conditions, including rheumatoid arthritis, low-grade systemic inflammation, and periodontitis [31]. This inflammatory phenotype may be driven by its unique ability to encode a superoxide reductase, which provides resistance to host-derived reactive oxygen species (ROS) produced during inflammation, and may even exploit these ROS to its advantage [32]. In mouse models, colonization with *P. copri* has been shown to exacerbate inflammation, as evidenced by increased severity in a colitis model induced by dextran sulfate sodium [33]. Additionally, in vitro studies reveal that *P. copri* can stimulate the production of key pro-inflammatory cytokines, including IL-6, IL-23, and IL-17, which are associated with Th17-mediated immune responses [34]. These findings highlight *P. copri*’s potential role as a driver of inflammation across a range of conditions.

To explore causality and demystify the mechanisms of *P. copri*-mediated MASLD progression, we populated *P. copri* into the gastrointestinal tracts of mice through oral gavage. Despite similar food intake quantities, *P. copri*-colonized mice on a high fat diet exhibited hepatic steatosis symptoms relative to control mice on a high fat diet: increased body weight and elevated hepatic triglycerides. The liver is central to regulating lipid balance throughout the body by controlling various processes related to fatty acids (FAs): their uptake, synthesis, oxidation, and export [35]. These FAs are integrated into triglycerides, the main form of stored fat within the liver [35]. In a non-disease state, the liver can export these triglycerides in the form of very-low-density lipoprotein (VLDL). However, under conditions of metabolic dysfunction, such as insulin resistance, the liver’s ability to package and export triglycerides as VLDL is impaired, leading to increased triglycerides within hepatocytes [36]. The presence of chronic triglyceride-rich droplets within liver cells is diagnosed as steatosis, which progresses to later stages of MASLD such as advanced fibrosis and liver cancer. These findings support our hypothesis that *P. copri* leads to the progression of MASLD towards an advanced fibrosis-like phenotype.

To investigate the molecular mechanisms driving MASLD progression through hepatic steatosis, we analyzed the expression levels of three key lipid metabolism genes from liver samples: carnitine palmitoyltransferase 1 (*Cpt1*), diacylglycerol acyltransferase (*Dgat*), and adipose triglyceride lipase (*Atgl*). Each of these genes plays a critical role in maintaining lipid homeostasis by regulating fatty acid oxidation, triglyceride synthesis, and lipolysis, respectively. While we saw no significant differences in *Dgat* expression, there were significant decreases in *Cpt1* and *Atgl* levels in high-fat *P. copri* mice compared with high-fat control mice.

*Cpt1* is essential for the transport of long-chain fatty acids into the mitochondria, where they undergo beta-oxidation to produce energy [37]. As the rate-limiting enzyme for fatty acid oxidation, *Cpt1* facilitates the conversion of fatty acyl-CoA to acyl-carnitine, enabling fatty acid entry into mitochondria [37]. Inhibition and downregulation of *Cpt1* prevents the entry of fatty acids into the mitochondria for oxidation, resulting in hepatic lipid accumulation and fostering the progression of liver steatosis and metabolic dysfunction [37]. Meanwhile, *Atgl* initiates the breakdown of stored triglycerides into free fatty acids and glycerol through lipolysis, which mobilizes fat stores for energy utilization [38]. Reduced *Atgl* expression hinder lipolysis, resulting in triglyceride accumulation in the liver and exacerbating the hepatic fat storage imbalance associated with MASLD.

In addition to disrupted lipid metabolism, *P. copri* may drive MASLD progression by promoting inflammation and scarring within liver tissue through effects on the gut barrier. This barrier is a complex defense system comprising a protective mucus layer, with an outer, thinner layer and an inner, thicker layer, followed by a layer of epithelial cells held together by tight junctions [39]. Tight junctions are proteins, mainly claudins (*Cldn*), occludin (*Ocln*), and zonula occludens-1 (*Zo*), that maintain gut epithelial integrity and selectively control the passage of substances through the barrier [40,41]. This barrier structure prevents microbial pathogens from invading intestinal epithelial cells while allowing nutrient absorption, providing a balance essential for intestinal health [41]. Disruptions in the gut microbial communities can increase gut permeability, allowing microbial products and toxins to escape into the bloodstream, thereby triggering systemic inflammation that can extend to the liver [41].

Our findings reveal that *P. copri* colonization in mice on a HF diet results in a significant reduction in the expression of key tight junction genes—*Ocln*—at the hepatic flexure, a region of the colon near the liver. With weakened tight junctions due to altered microbial composition, the gut barrier’s selective permeability may be compromised, allowing bacterial components, such as lipopolysaccharides, to cross from the gut lumen into circulation [42]. This translocation can trigger hepatic inflammation by activating immune receptors, like Toll-like receptor 4, in liver cells, promoting an inflammatory cascade that may lead to fibrosis and liver tissue scarring.

While our findings provide strong evidence linking *P. copri* to advanced fibrosis and MASLD progression in both human and mouse models, there are important limitations to acknowledge. First, the small sample size in our human cohort, particularly among patients with advanced fibrosis, may constrain the generalizability of these results. On a similar note, we relied on FibroScan rather than liver histology to perform the diagnosis of hepatic fibrosis, but this has become a more widely accepted method for detecting hepatic fibrosis [43]. Additionally, the absence of long-term outcome data limits our ability to fully evaluate how the observed microbial changes influence the trajectory of MASLD progression in both humans and mice. Notably, we did not observe significant differences in fibrosis between groups, likely due to the extended time required for fibrosis to develop. Collecting longer-term data in future studies would address this limitation and provide a more comprehensive understanding of the relationship between *P. copri* and fibrosis progression.

## 5. Conclusions

Overall, our findings revealed a significant association between *P. copri* abundance in the gut and advanced fibrosis in MASLD patients. In turn, we provided experimental evidence that *P. copri* exacerbates hepatic steatosis, disrupts lipid metabolism, and impairs gut barrier integrity in a diet-induced obesity model. Collectively, these results position *P. copri* as more than a marker for disease severity, but an active driver of MASLD progression. By targeting specific microbial signatures, such as *P. copri*, in conjunction with metabolic interventions, there is potential to develop novel therapeutic strategies that prevent MASLD progression toward advanced fibrosis and cirrhosis. Future research should explore the mechanistic pathways of *P. copri* and its metabolites to better understand its role in MASLD and the broader implications of the gut–liver axis in chronic liver disease.

## Figures and Tables

**Figure 1 nutrients-17-02145-f001:**
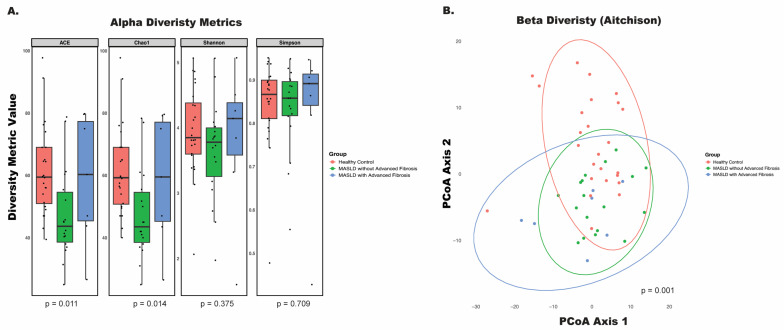
Microbial diversity in MASLD patients with and without advanced fibrosis. (**A**) Alpha diversity metrics (ACE, Chao1, Shannon, Simpson) measured in stool samples from three groups: healthy controls (red), MASLD patients without advanced fibrosis (green), and MASLD patients with advanced fibrosis (blue). Significant differences in ACE and Chao1 diversity indices were observed, indicating altered microbial richness in MASLD with advanced fibrosis (*p* = 0.011 and *p* = 0.014, respectively). (**B**) Beta diversity analysis using the Aitchison distance metric, visualized through principal coordinates analysis (PCoA), showing significant clustering of microbial communities across the groups (*p* = 0.001, PERMANOVA).

**Figure 2 nutrients-17-02145-f002:**
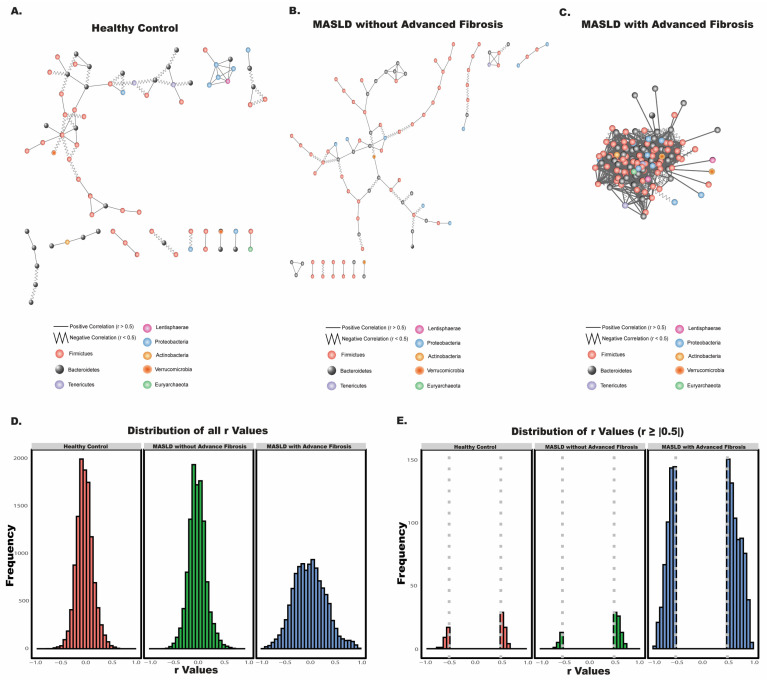
Bacterial co-occurrence interactions in MASLD patients with and without advanced fibrosis. (**A**–**C**) Co-occurrence networks showing microbial associations in (**A**) healthy controls, (**B**) MASLD patients without advanced fibrosis, and (**C**) MASLD patients with advanced fibrosis. Nodes represent bacterial taxa, colored by phylum, and edges represent significant correlations (r ≥ |0.5|) between taxa. (**D**) Distribution of all bacterial correlation (r) values in each group. (**E**) Distribution of correlation (r) values filtered by significant correlations (r ≥ |0.5|), highlighting differences in strong microbial associations across disease groups.

**Figure 3 nutrients-17-02145-f003:**
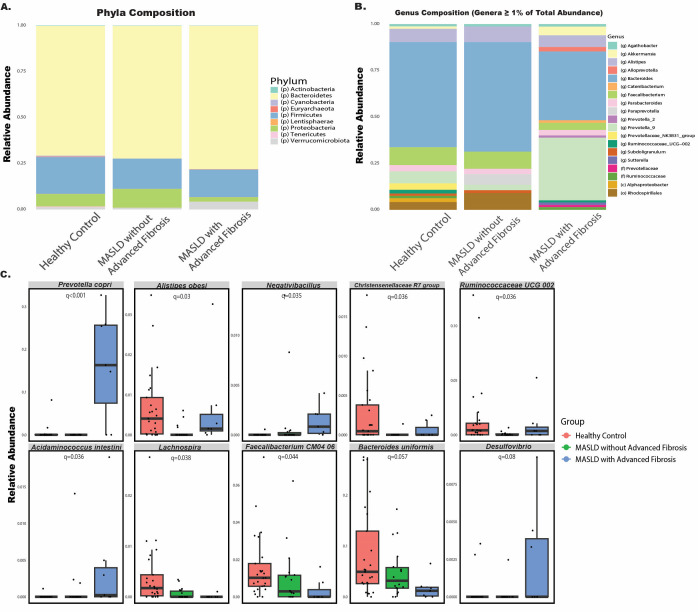
Taxonomic composition differences highlight elevated *P. copri* abundance in MASLD patients with advanced fibrosis. (**A**) Stacked bar plots showing the relative abundance of bacterial phyla across healthy controls, MASLD patients without advanced fibrosis, and MASLD patients with advanced fibrosis. (**B**) Stacked bar plots displaying the relative abundance of bacterial genera (restricted to those constituting ≥1% of total abundance) within each group, indicating notable changes in genus composition as MASLD progresses. (**C**) Box plots of selected species with differential abundance across groups (q < 0.1), emphasizing the significantly higher abundance of *P. copri* in MASLD patients with advanced fibrosis and highlighting other genera with altered representation across the stages of MASLD progression.

**Figure 4 nutrients-17-02145-f004:**
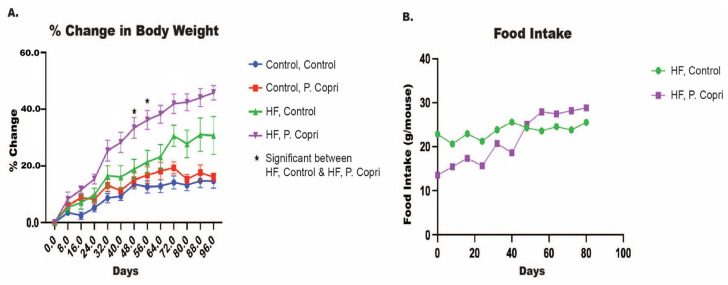
Effect of *P. copri* colonization on body weight and food intake in high-fat diet mice. (**A**) Median percent change in body weight over time with 95% confidence intervals for mice on a high-fat (HF) or control diet, with or without *P. copri* colonization. Mice in the HF diet group gavaged with *P. copri* (purple) showed significantly greater body weight gain compared to HF diet controls (green) at multiple time points (*p* < 0.01). (**B**) Average food intake (g/mouse) over time for HF diet groups. *n* = 8 per group.

**Figure 5 nutrients-17-02145-f005:**
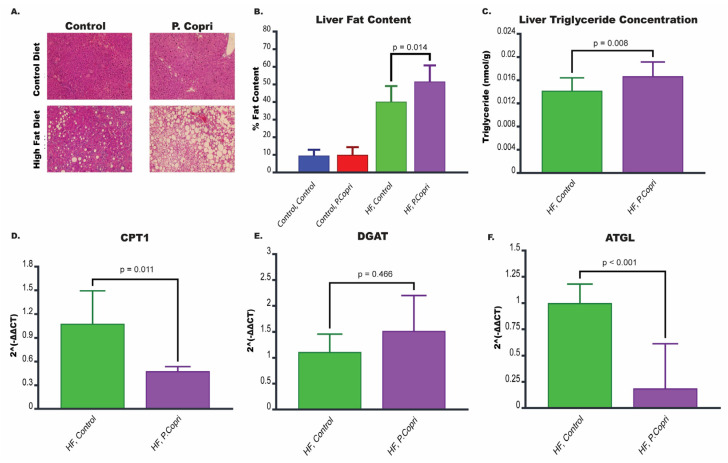
Impact of *P. copri* colonization on hepatic steatosis, triglyceride content, and lipid metabolism gene expression in high-fat diet mice. (**A**) Representative hematoxylin and eosin (H&E) stained liver sections showing fat accumulation in mice on a standard control diet or high-fat (HF) diet, with or without *P. copri* colonization. White areas indicate lipid accumulation in the liver, which was most pronounced in the HF diet *P. copri* group. (**B**) Quantification of liver fat content for HF *P. copri* group compared to the HF control group using ImageJ. (**C**) Triglyceride concentrations in the liver between HF diet colonized with *P. copri* and HF control mice. (**D**–**F**) Expression of genes involved in lipid metabolism for the HF *P. copri* group compared to the HF control group. These genes include (**D**) carnitine palmitoyltransferase 1 (*Cpt1*), (**E**) diacylglycerol acyltransferase (*Dgat*), and (**F**) adipose triglyceride lipase (*Atgl*). *n* = 8 per group.

**Figure 6 nutrients-17-02145-f006:**
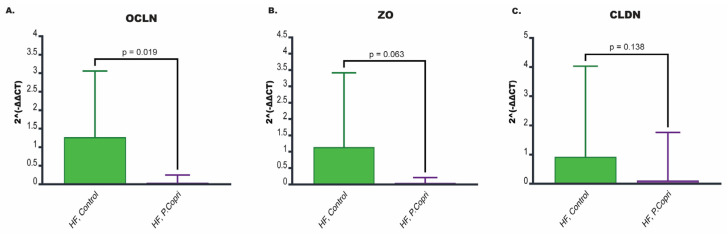
Effect of *P. copri* colonization on tight junction gene expression in the hepatic flexure of high-fat diet mice. (**A**–**C**) Expression of genes involved in tight junction integrity acquired from the hepatic flexure of the HF *P. copri* group and the HF control group. These genes include (**A**) occludin (Ocln), (**B**) zonula occludens-1 (Zo), and (**C**) claudins (Cldn). *n* = eight per group.

**Table 1 nutrients-17-02145-t001:** Patient demographic.

	Control (*n* = 25)	MASLD without Advanced Fibrosis (*n* = 18)	MASLD with Advanced Fibrosis (*n* = 7)	*p*-Value
Age (yr) (SD)	35.7 (3.5)	57.2 (16.0)	67.4 (8.0)	<0.001
Male (%) (*n* = 50)	52% (*n* = 13)	83% (*n* = 15)	100% (*n* = 7)	0.018
Charlson Comorbidity Index (SD)	N/A	3.9 (2.9)	5.7 (1.5)	0.06
Caucasian (%) (*n* = 17)	32% (*n* = 8)	33.3% (*n* = 6)	42.9% (*n* = 3)	0.484
African American (%) (*n* = 15)	32% (*n* = 8)	33.3% (*n* = 6)	14.3% (*n* = 1)	
Hispanic (%) (*n* = 8)	8% (*n* = 2)	27.8% (*n* = 5)	14.3% (*n* = 1)	
Asian (%) (*n* = 5)	16% (*n* = 4)	0% (*n* = 0)	14.3% (*n* = 1)	
Other/Unknown (%) (*n* = 5)	12% (*n* = 3)	5.6% (*n* = 1)	14.3% (*n* = 1)	

## Data Availability

Raw 16s rRNA sequence data were deposited under National Center for Biotechnology Information Bioproject PRNA542724.

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
