# Peer review of "The Association Between Prevotella copri and Advanced Fibrosis in the Progression of Metabolic Dysfunction-Associated Steatotic Liver Disease"

_nutrients, 2025, doi:10.3390/nu17132145_

Round 1

Reviewer 1 Report

Comments and Suggestions for Authors

General comments:

The manuscript is interesting and very well written. The high degree of similarity index obtained after analysis by Itenticate software corresponds mainly to coincidences with a manuscript deposited in a repository by the same authors and unpublished, so it is not relevant. What is somewhat strange is that this manuscript has been submitted to a special issue on the influence of the intestinal microbiota on obesity. It is not deduced that the subjects who have participated in the study are obese, since only their racial characteristics and MASLD progression are shown.

Although MASLD is a pathology closely related to obesity, this is hardly mentioned in the manuscript and the relationship between MASLD and obesity should be more clearly stated in the introduction to better justify the inclusion of this manuscript in this Special issue.

Specific comments:

Formats are not current: The journal name (Nutrients) was omitted on the first page, and the references are not between brackets in the text.

Please, could you explain what is an “anaconda environment”?

Table headings should be placed upon the table, not lower. Please delete the redundant “table 1”, and dark funds, according to editorial´s guidelines.

Figure 3 names are too low size.

Page 13: genus names should be written in italics. The same “P. copri” in page 15.

References are not according MDPI´s guidelines.

Author Response

Comment 1: The manuscript is interesting and very well written. The high degree of similarity index obtained after analysis by Itenticate software corresponds mainly to coincidences with a manuscript deposited in a repository by the same authors and unpublished, so it is not relevant. What is somewhat strange is that this manuscript has been submitted to a special issue on the influence of the intestinal microbiota on obesity. It is not deduced that the subjects who have participated in the study are obese, since only their racial characteristics and MASLD progression are shown.

Although MASLD is a pathology closely related to obesity, this is hardly mentioned in the manuscript and the relationship between MASLD and obesity should be more clearly stated in the introduction to better justify the inclusion of this manuscript in this Special issue.

Response 1: Yes, we agree. While MASLD is highly related to obesity it should be more clearly stated. So, we edited the introductions to reflect that.

Specific comments:

Comment 2: Formats are not current: The journal name (Nutrients) was omitted on the first page, and the references are not between brackets in the text.

Response 2: Thank you, we have made this change

Comment 3: Please, could you explain what is an “anaconda environment”?

Response 3: Yes, we have updated that. Anaconda is a python environment.

Comment 4: Table headings should be placed upon the table, not lower. Please delete the redundant “table 1”, and dark funds, according to editorial´s guidelines.

Response 4: We have made those corrections.

Comment 5: Figure 3 names are too low size.

Response 5: We have edited figure 3 to make the labels larger

Comment 6: Page 13: genus names should be written in italics. The same “P. copri” in page 15.

Response 6: We have updated this.

Comment 7: References are not according MDPI´s guidelines.

We have updated this.

Reviewer 2 Report

Comments and Suggestions for Authors

The research paper entitled "The Association Between Prevotella copri and Advanced Fibrosis in the Progression of Metabolic Dysfunction-Associated Steatotic Liver Disease" is very clear, well written and organized.

However, the paper needs to be improved.

Indeed only 25 patients with MASLD and 25 healthy controls were recruited. On these samples, statistixcal data are realized with P<0.05. As this study can be considered as a pre-clinical study, I think that statistical analyses must also be presented at P<0.01. The data will be therefore more reliable and this will permit to improve the quality of the discussion which must take in consideration the small sample considered.

In mice, the total number of mice is indicated, but the number of mice per group is not indicated in the figure legends concerned. This must be added. Similarly, statistical analysis at P<0.01 must also be presented.

These modifications are mandatory and I think that they will increase the quality of the manuscript especially the discussion but also the conclusion.

P. copri is more abundant in MASLD patients with advanced fibrosis

Author Response

Comment 1: Indeed only 25 patients with MASLD and 25 healthy controls were recruited. On these samples, statistixcal data are realized with P<0.05. As this study can be considered as a pre-clinical study, I think that statistical analyses must also be presented at P<0.01. The data will be therefore more reliable and this will permit to improve the quality of the discussion which must take in consideration the small sample considered.

Response 1:Thank you for this comment. We agree with the reviewer. We have gone through the paper and listed each individual p-values, therefore readers can discern which ones are below 0.05 and which are 0.01. As for q-values, q-values are calculated after adjusting for false discovery and in figure 3 we listed each q-values of each bacteria so that readers can more easily determine the data’s reliability. The discussion we focused on for the human study was on those that were on those that had a p-value of <0.01.

Comment 2: In mice, the total number of mice is indicated, but the number of mice per group is not indicated in the figure legends concerned. This must be added. Similarly, statistical analysis at P<0.01 must also be presented.

Response 2: We have added this to the figure legends and changed the figures to show the p-values that are less than 0.01, which was updated in figure 4. We also only discussed those results that were less than 0.01.